# Investigation of Slow Eutectoid Element on Tensile Properties and Superplasticity of a Forged SP700 Titanium Alloy

**Dong Han [1], Yongqing Zhao [1,2,\*], Weidong Zeng [1] and Junfeng Xiang [1,3,\*]**

[1]  School of Materials Science and Engineering, Northwestern Polytechnical University, Xi'an 710072, China; handongnin@mail.nwpu.edu.cn (D.H.); zengwd@nwpu.edu.cn (W.Z.)
[2]  Northwest Institute for Nonferrous Metal Research, Xi'an 710016, China
[3]  School of Civil Aviation, Northwestern Polytechnical University, Xi'an 710072, China
\*  Correspondence: trc@c-nin.com (Y.Z.); xiangjf@nwpu.edu.cn (J.X.)

**Abstract:** The tensile properties and superplasticity of a forged SP700 alloy with slow eutectoid element (1.5%Cr) addition were investigated in the present paper. The results of the microstructures showed that slow eutectoid element Cr has a significant influence on stabilizing the β phase and the SP700Cr alloy showed a uniform duplex and completely globular microstructure after annealing at 820 °C for 1 h and aging at 500 °C for 6 h. The results of the tensile tests showed that the yield strength, ultimate tensile strength and elongation of the alloy with optimized microstructure were 1312 MPa, 1211 MPa and 10% at room temperature, and the elongation was achieved to 1127% at 770 °C. Compared with that of the SP700 alloy, the strain rate sensitivity of the SP700Cr alloy showed a higher value. The microstructures after elevated temperature tensile tests showed that the higher density of dislocations and twins exists in SP700 alloy and the lower density of dislocations favor distribution in SP700Cr alloy. Based on the above results, the tensile properties and superplasticity of the forged SP700 alloy with 1.5% Cr addition was analyzed. In addition, microstructure characteristics were investigated by the TEM and EBSD technologies.

**Keywords:** SP700 alloy; slow eutectoid element; microstructure; tensile properties; superplasticity

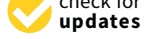



## 1. Introduction

Titanium alloys have been widely used in the aerospace, automobile, chemical and other industries due to its desirable properties of higher strength, ductility and superplasticity, in which, SP700 alloy (a kind of excellent β-rich α + β titanium alloy) is attracting increasing attention for its excellent superplasticity [1–6]. Superplasticity refers to the capability of high tensile elongation before failure of polycrystalline materials, which is usually characterized by low flow stress and high plastic strain. In past years, the superplasticity of titanium alloys has been widely investigated [1,2,4,7]. However, the temperature of superplasticity deformation of titanium alloys is generally relatively high and it results in significant difficulties in many practical applications of the alloys. For this purpose, it is desirable to find methods to enhance the superplasticity at lower temperature or at higher strain rate. As known, refining the grain structures is favorable for superplasticity at elevated temperature, in which the equiaxed recrystallized grain with high angle grain boundaries or the non-recrystallized grain are two typical microstructures to obtain excellent superplasticity [8–10]. To achieve this kind of grain refinement, severe plastic deformation (equal-channel angular pressing and high-pressure torsion, etc.) as an effective method have been widely used [3,11,12]. However, this method would increase the cost of processing. Meanwhile, the amount of material that can be processed by the method is also limited. Thus, some researchers concentrate on the methods to improve the superplasticity by optimizing the chemical composition through adding alloy elements that can control grain size or using β stabilizing elements that can promote the superplasticity at low temperature [13].

The β phase stable elements can reduce the transition temperature of α→β, and they can be divided into isomorphic elements and eutectoid ones. The β eutectoid elements can be divided into fast and slow eutectoid elements. Slow eutectoid elements include Mn, Fe, Cr, etc. the strengthening effect of these elements is great and stable β phase ability. At the same time, they make the eutectoid reaction of the β phase slow down, and the reaction is too late at the general cooling rate, resulting in solid solution strengthening of the alloy. Therefore, based on the previous research on the neutral element Zr of SP700 alloy, it is necessary to further clarify the role of eutectoid elements of β phase. The effect of slow eutectoid elements on the properties of the alloy should be studied [14].

As a slow eutectoid element of β phase, Cr can decrease the transformation temperature of β phase and the superplasticity temperature. Furthermore, Cr is prone to be abundant and inexpensive without compromising the performance requirement [15–17]. Meanwhile, heat treatment, as a useful method to optimize the microstructure and mechanical properties of titanium alloys, has been widely used to control the size and content of the α lamellar phase, the phase ratio of α to β phases, and the morphology of the β phase [18,19]. Thus, an optimized microstructural morphology and an adjusted volume fraction ratio of α phase and β phase can be obtained by combining the above two typical methods.

In the present study, a series of heat treatments is firstly performed in the SP700 alloy to investigate the microstructural evolution. Then the mechanical properties performed at room temperature with different microstructures are studied by uniaxial tensile tests. Meanwhile, the superplasticity behaviors with optimized microstructure are also studied at elevated temperature. To explore the mechanical properties and superplasticity behaviors and microstructure, the corresponding data are analyzed and studied in detail.

## 2. Materials and Experimental Procedures

The materials used in this paper were forged SP700 and SP700Cr titanium alloys. The selected sample was a cylindrical sample, with a diameter of 80 mm and a height of 120 mm. It was forged to 96 mm with compression reduction of 20%. The forging temperature was set as 845 °C. The final sample was a cylinder with a diameter of 88.9~95.2 mm with a middle bulge. Subsequent tests were carried out after tempering at 400 °C. The chemical composition of the alloys is shown in Table 1. The designed heat treatment schedules are listed in Table 2.

**Table 1.** Chemical composition of the forged SP700 and SP700Cr alloys.

| Alloys | Chemical Compositions (wt.%) | | | | | | | | | |
|--------|------|------|------|------|------|------|-------|-------|-------|-------|
| | **Ti** | **Al** | **V** | **Mo** | **Fe** | **Cr** | **C** | **N** | **O** | **H** |
| SP700 | Bal. | 4.61 | 3.15 | 2.03 | 1.51 | 0 | 0.008 | 0.006 | 0.093 | 0.002 |
| SP700Cr | Bal. | 4.52 | 3.09 | 1.98 | 1.44 | 1.18 | 0.007 | 0.005 | 0.087 | 0.002 |

**Table 2.** Heat treatment schedules of the forged SP700 and SP700Cr alloys.

| No. | Heat Treatment Schedules |
|-----|--------------------------|
| AC710 | 710 °C 1 h, Air Cooling (AC) |
| AC800 | 800 °C 1 h, AC |
| AC820 + AC500 | 820 °C 1 h AC + 500 °C 6 h AC |

The microstructures of the alloys were observed on an optical microscope (OM, Olympus GX71; Olympus Corporation, Tokyo, Japan). The sample for OM observation was firstly mechanically polished to a mirror-like surface and then etched with a solution (5% HF + 10% $HNO_3$ + 85% $H_2O$). Tensile tests were conducted on an Instron 5869 testing machine and performed at room temperature and elevated temperatures (730 °C, 750 °C, 770 °C and 790 °C) and different strain rates ($1 \times 10^{-2}$ s$^{-1}$, $5 \times 10^{-3}$ s$^{-1}$, $1 \times 10^{-3}$ s$^{-1}$, $5 \times 10^{-4}$ s$^{-1}$ and $1 \times 10^{-4}$ s$^{-1}$). The tensile sample was designed as a dog-bone shape

with a gauge size of diameter of φ 5 mm and length of 25 mm, and the gauge length was machined with the direction parallel to the length of the forged plate. The tensile tests were conducted three times at least and the typical curves are shown in the present study. The tensile tests were carried out according to the Chinese National Standards (GB/T4338-2006). The morphologies of the sample after superplasticity deformation were observed by the transmission electron microscope (TEM, JEM-F200; JEOL, Tokyo, Japan). Specimens for TEM observation were prepared by grinding to a thickness of about 20 μm, punched out 3 mm in diameter, and then thinned to a thickness of electron transparency on an argon-ion milling system (RES101; Leica, Wetzlar, Germany). The specimens were electropolished for the EBSD observations. The EBSD tests were conducted on a TESCAN scanning electron microscope (TESCAN, Brno, Czech Republic), and the step size of the EBSD test was set as 0.3 μm. The data analysis was carried out based on the HKL-Channel 5 software.

## 3. Results

### 3.1. Microstructures

Figure 1 shows the optical images of the forged SP700 alloy and the alloys after different heat treatments. It can be clearly seen from Figure 1a,b that the forged SP700 alloy is consist of two contrast phases arranged at intervals in strip shape, i.e., the primary α phase in white and the β phase in black. The primary α phase is non-uniformly distributed on the β phase matrix. It can be seen that the α phase is still in a strip shape in the SP700 alloy after AC710 (Figure 1c,d). However, the number of primary α phase decreases and the number of β phase increases, which may be because the primary α phase is gradually dissolved and finally forms the unstable solid solution during the heat treatment. After AC800 (Figure 1e,f), almost all the primary α phases were transformed into the short strip and/or spherical β phases in the SP700 alloy. The morphology of the SP700 alloy becomes the more equiaxed and uniform. After AC820 + AC500 (Figure 1g,h), the SP700 alloy presents the uniform duplex microstructures, i.e., the α phases distributed in a dispersed way in the matrix (primarily consisting of β phases). The α phases in SP700 alloy are in short strip and/or small spherical shapes.

Figure 2 shows the optical images of the forged SP700Cr alloy after different heat treatments. It can be seen from Figure 2a,b that the forged SP700Cr alloy also consisted of primary α phase (white) and β phase (black) arranged at intervals in strip shape. As compared with that of forged SP700 alloy, the length of the α and β phases become smaller and the distribution of the two phases become more uniform. It can also be observed that the volume fraction of the β phase increases in the forged SP700Cr alloy. After AC710 (Figure 2c,d), it was found that the long lamellar α phase changed into the short ones although it was still in a strip shape in the SP700Cr alloy. Meanwhile, the number of the primary α phase increases and microstructure of the SP700Cr alloy becomes more uniform. After AC800 (Figure 2e,f), almost all the primary α phases were transformed into the short strip and/or spherical β phases in the SP700 alloy. The morphology of the SP700 alloy becomes the more equiaxed and uniform. After AC820 + AC500 (Figure 2g,h), the SP700 alloy presents uniform duplex microstructures. The α phases are in short strip and/or small spherical shapes and distributed in a dispersed way in the matrix (primarily consisting of β phases), i.e., the SP700Cr alloy displays a much more obviously equiaxed and globular microstructure.

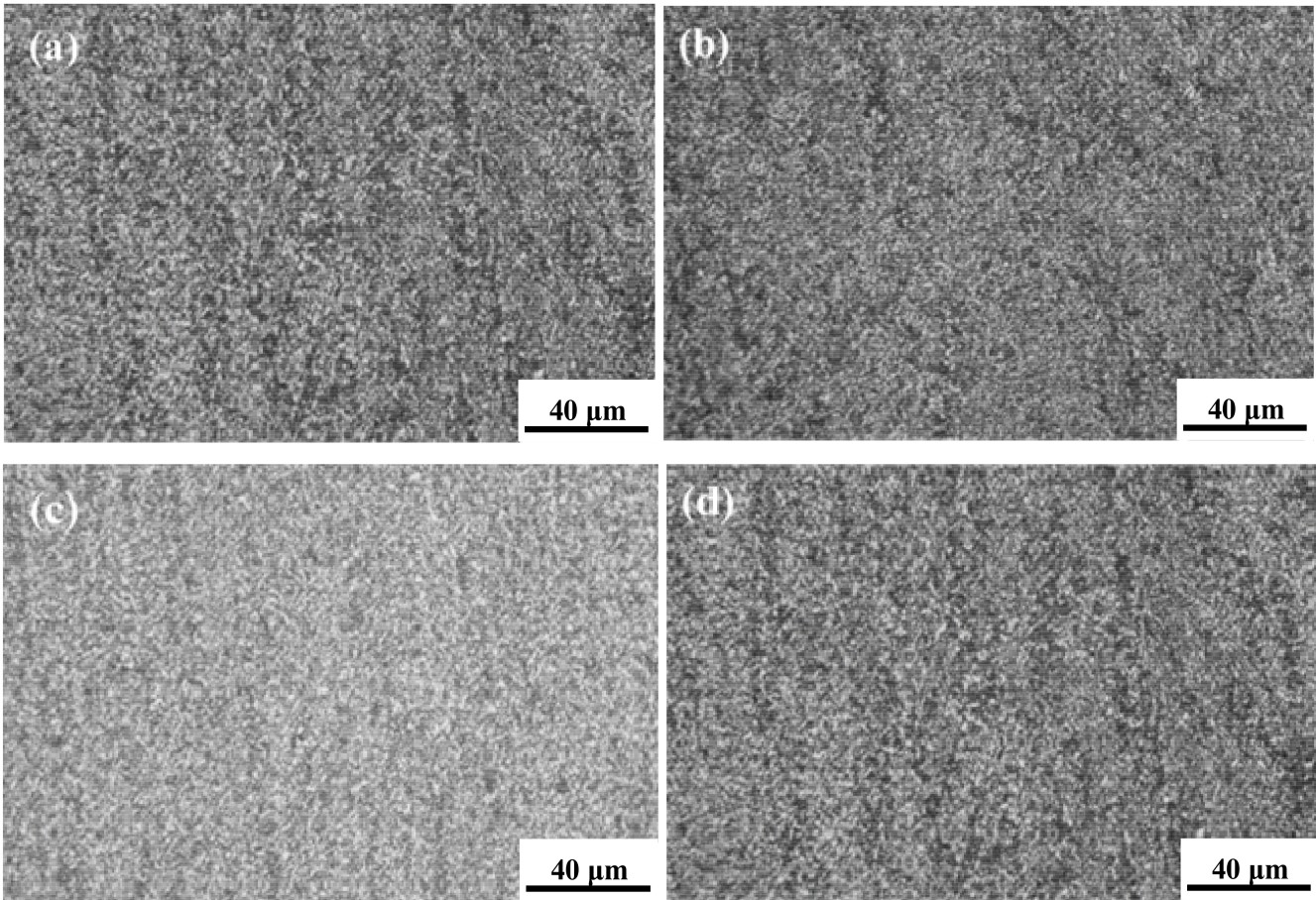

**Figure 1.** Optical images of the forged (**a**) SP700, (**b**) SP700 alloy after AC710, (**c**) SP700 alloy after AC800 and (**d**) SP700 alloy after AC820 + AC500.

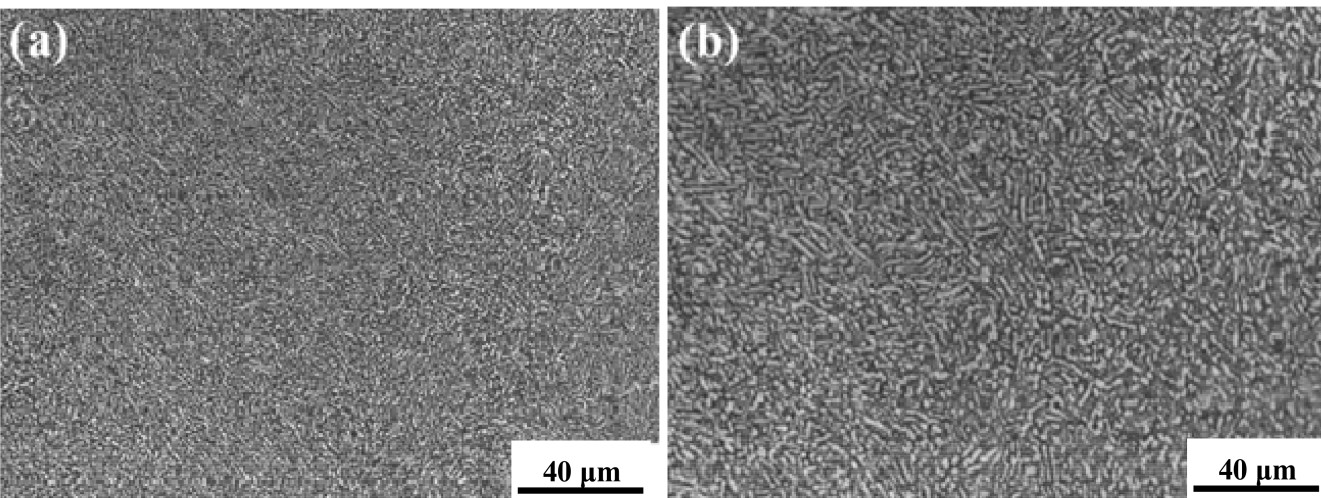

**Figure 2.** *Cont.*

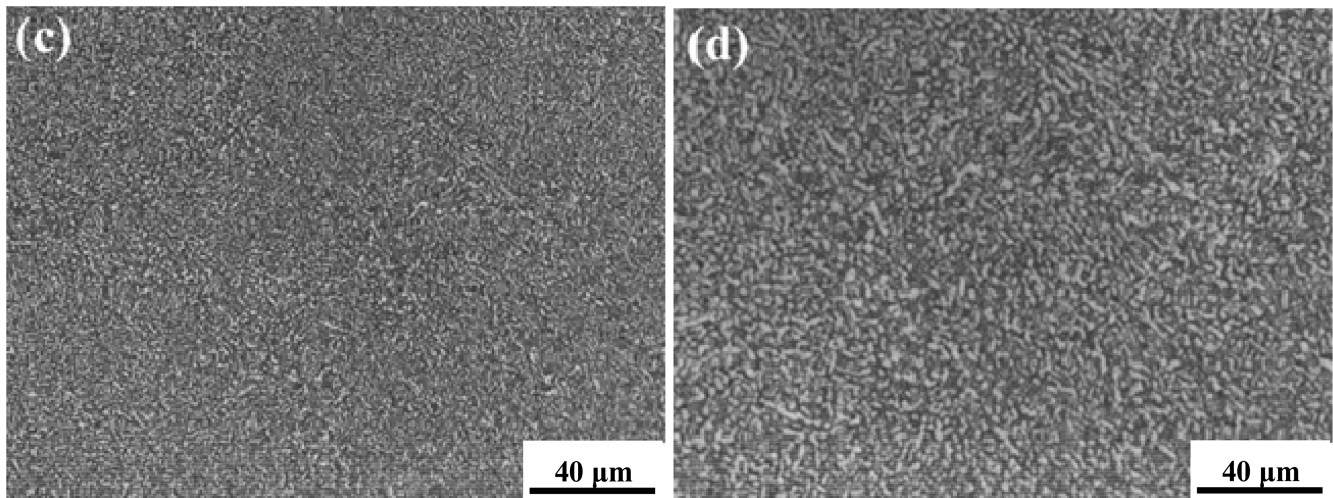

**Figure 2.** Optical images of the forged (**a**) SP700Cr, (**c**) SP700Cr alloy after AC710, (**e**) SP700Cr alloy after AC800 and (**g**) SP700Cr alloy after AC820 + AC500.

### 3.2. Tensile and Superplasticity Properties

Figure 3 gives the tensile properties at room temperature of both alloys after different heat treatments. It can be seen that both alloys exhibit the highest tensile strength and lowest elongation after AC820 + AC500 than that after AC710 and AC800. After AC820 + AC500, the SP700Cr alloy shows superior tensile strength and unchangeable elongation as compared with that of SP700 alloy. However, after AC710 and AC800, the strength and elongation of the SP700Cr alloy are nearly the same as that of SP700 alloy. It is particularly necessary to point out that the degraded ductility following the AC820 + AC500 treatment is an important limitation for an engineering application despite the increase in strength. The relevant information must be fully considered in an engineering application.

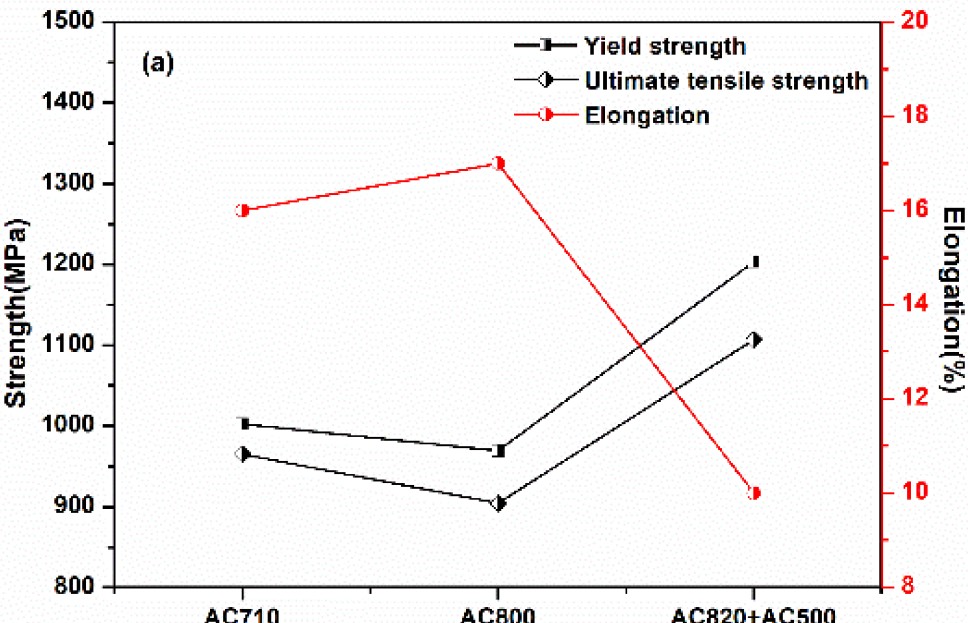

**Figure 3.** *Cont.*

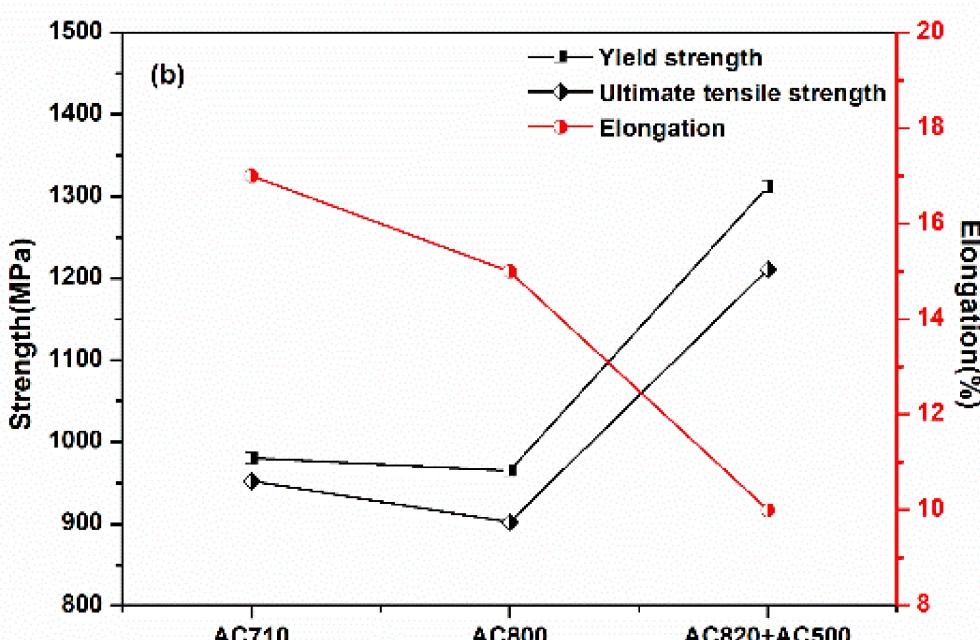

**Figure 3.** Tensile properties of forged (**a**) SP700 [14] and (**b**) SP700Cr at room temperature.

Figure 4 displays the flow stress–strain curves at elevated temperatures (730~790 °C) of the both alloys after AC820 + AC500 at a strain rate of $1 \times 10^{-3}$ s$^{-1}$. It can be seen that the flow stress of both alloys decreases with increasing tensile temperature. As for tensile strain, it initially increases and then decreases with increasing tensile temperature for SP700 alloy and nearly have no change for SP700Cr alloy. From the flow stress–strain curves, it can be clearly seen that the superplasticity is achieved at 750 °C for SP700 alloy (1094%) and at all tensile temperatures for SP700Cr alloy (865–1127%), indicating that Cr is a favorable element to improve the superplasticity for the SP700 alloy. This superplasticity behavior is also clearly seen in Figure 5 before and after fracture at different tensile temperatures of the both alloys after AC820 + AC500, in which the tensile elongation (1127%) of the SP700Cr alloy is much higher than that of the SP700 alloy (1094%).

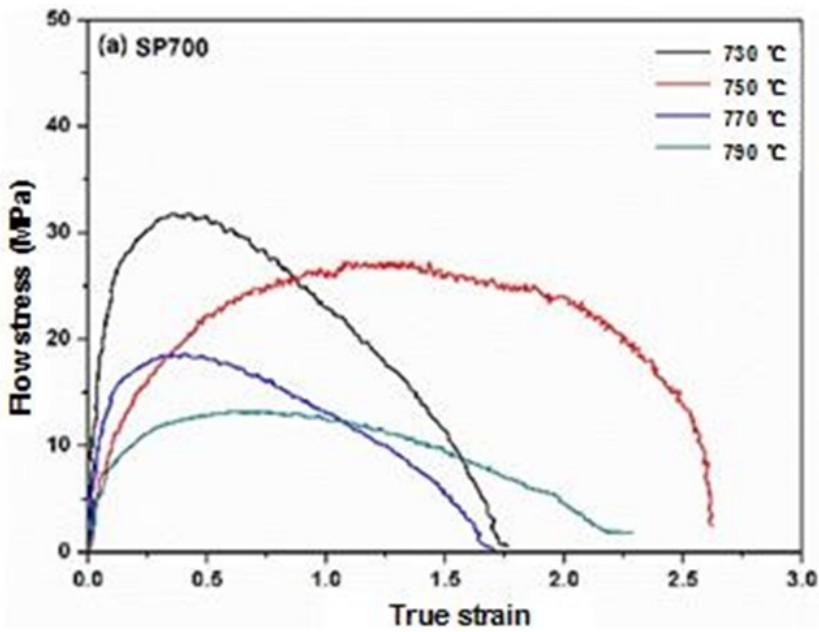

**Figure 4.** *Cont.*

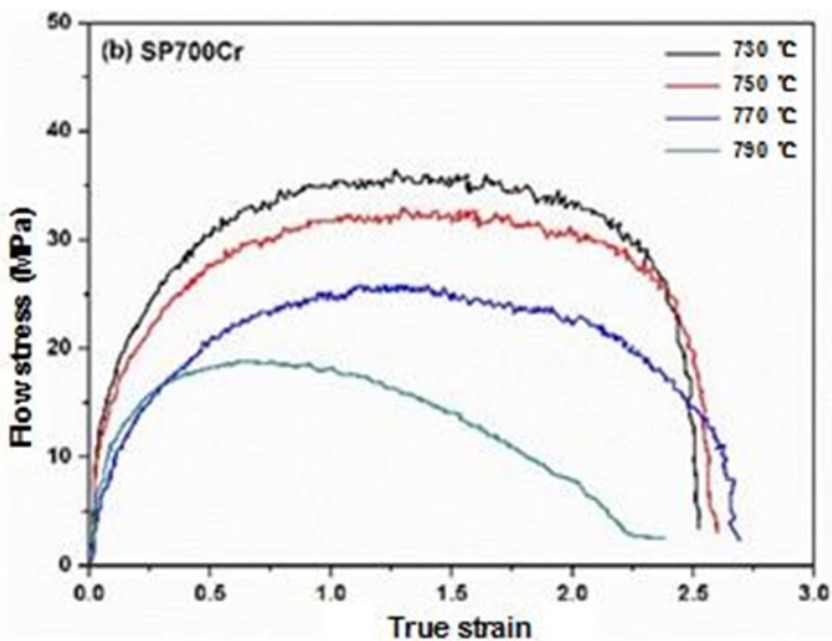

**Figure 4.** Flow stress-strain curves at elevated temperatures of forged (**a**) SP700 [14] and (**b**) SP700Cr alloys.

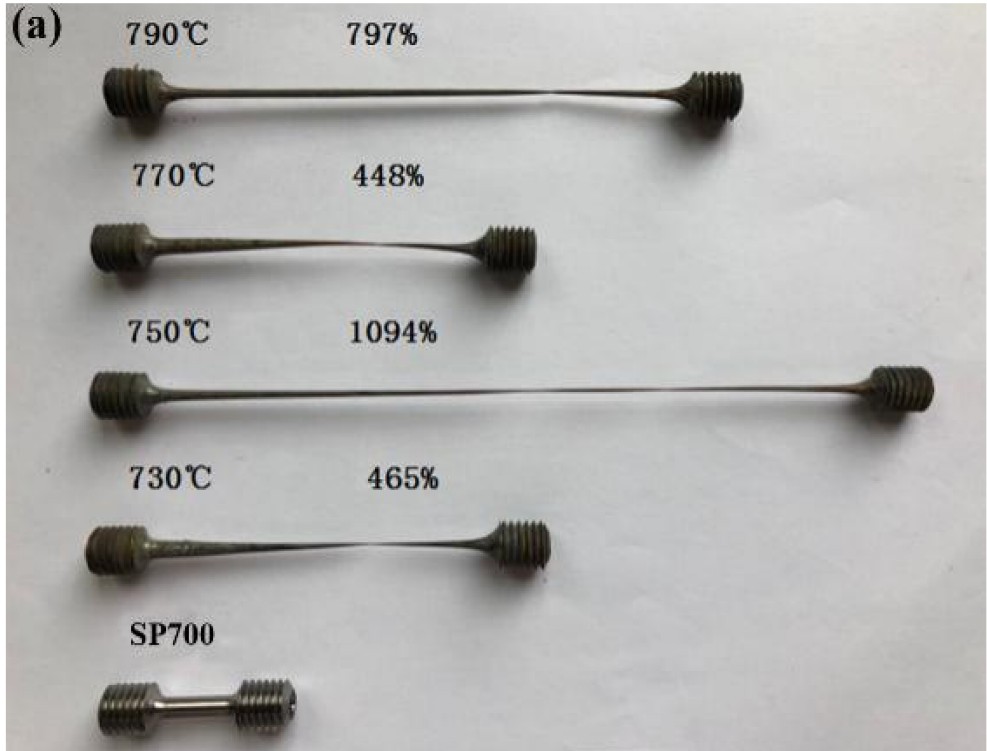

**Figure 5.** *Cont.*

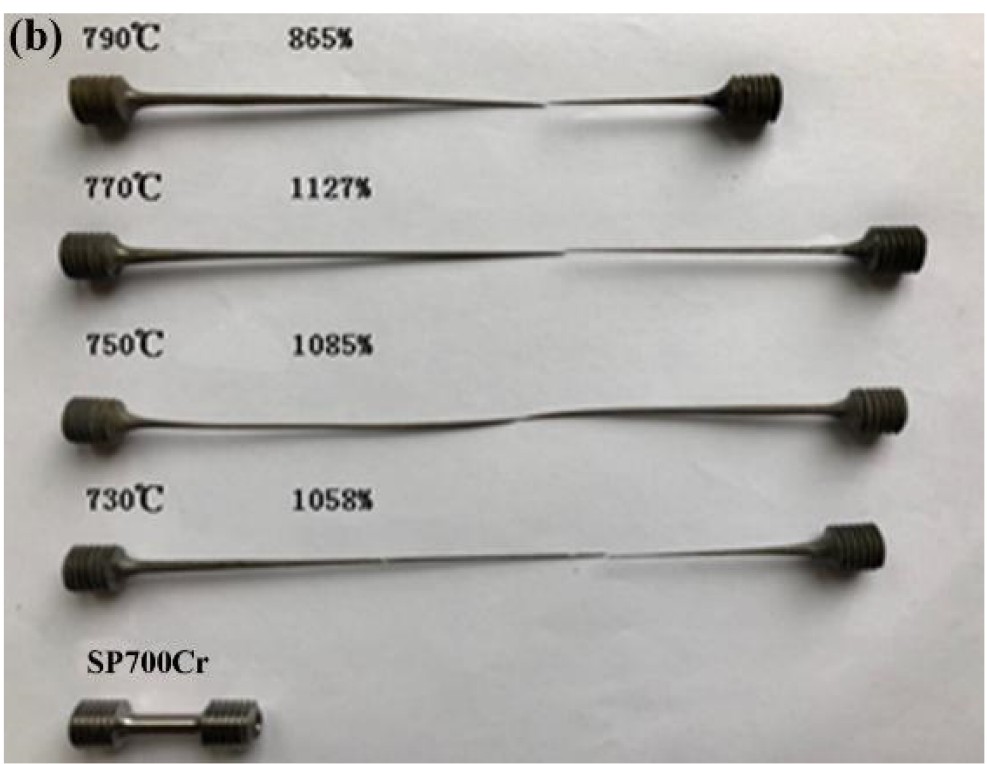

**Figure 5.** Images of the forged (**a**) SP700 [14] and (**b**) SP700Cr alloys before and after fracture at different tensile temperatures.

The work hardening rate ($\Theta$) is usually used to quantify the strain hardening response of the materials, which can be obtained from the stress–strain curves based on the following function [20]:

$$\Theta = \frac{\partial \sigma}{\partial \varepsilon} \tag{1}$$

where $\sigma$ and $\varepsilon$ are the flow stress and true strain, respectively. Figure 6 gives the work hardening rate curves of the present alloys obtained from the flow stress–strain curves in Figure 4. With increasing $\varepsilon$, it can be seen that the $\Theta$ values first drop rapidly (when $\varepsilon$ is lower than about 0.3) and then decrease slowly (when $\varepsilon$ is higher than about 0.3). The $\Theta$ values nearly overlap together when $\varepsilon$ is lower than about 0.3 and have significant difference when $\varepsilon$ is higher than about 0.3. When $\varepsilon$ is higher than about 0.3, the $\Theta$ value obtained at 750 °C is higher than others for SP700 alloy and the $\Theta$ value obtained at 730–770 °C is higher than 790 °C for SP700Cr alloy, which indicates that the SP700 alloy has higher work hardening ability when deformed at 750 °C and the SP700Cr alloy has higher work hardening ability when deformed at 730–770 °C. This work hardening ability is also identified by the flow stress–strain curves in Figure 4. This work hardening ability would decrease with increasing $\varepsilon$. The $\Theta$ value would become smaller than zero when $\varepsilon$ is higher than 0.4–1.25 for SP700 alloy and when $\varepsilon$ is higher than about 0.7–1.5 for SP700Cr alloy. This indicates that the role of strain softening becomes larger compared with that of strain hardening.

The strain rate sensitivity value m, which is measured from the analysis of stress–strain rate curves, is a function of the logarithmic values of the stress and strain rate for a given strain and temperature [21,22]:

$$m = \left. \frac{\partial ln\sigma}{\partial ln\dot{\varepsilon}} \right|_{\varepsilon, T} \tag{2}$$

where $\dot{\varepsilon}$ is the strain rate for a given value of strain $\varepsilon$ and temperature $T$. The derived $m$ values based on the tensile stress–strain curves at different strain rates and temperatures for the present alloys are displayed in Figure 7. It can be seen that the $m$ value of both alloys increases with increasing temperature. The optimal $m$ values are found at 750 °C in SP700 alloy and at 790 °C in SP700Cr alloy. As compared with that of SP700 alloy, the SP700Cr alloy has a larger $m$ value ranging from 0.35 to 0.53. For the polycrystalline metallic materials, the $m$ value is an important parameter to predict the superplasticity deformation mechanism. When the $m$ value reaches 0.5, grain boundary sliding should be the dominant deformation mechanism during the plastic deformation of metals and alloys. Therefore, the large $m$ value of 0.35–0.53 in SP700Cr alloy indicates that the grain boundary sliding is the dominant deformation mechanism contributing to the superplasticity.

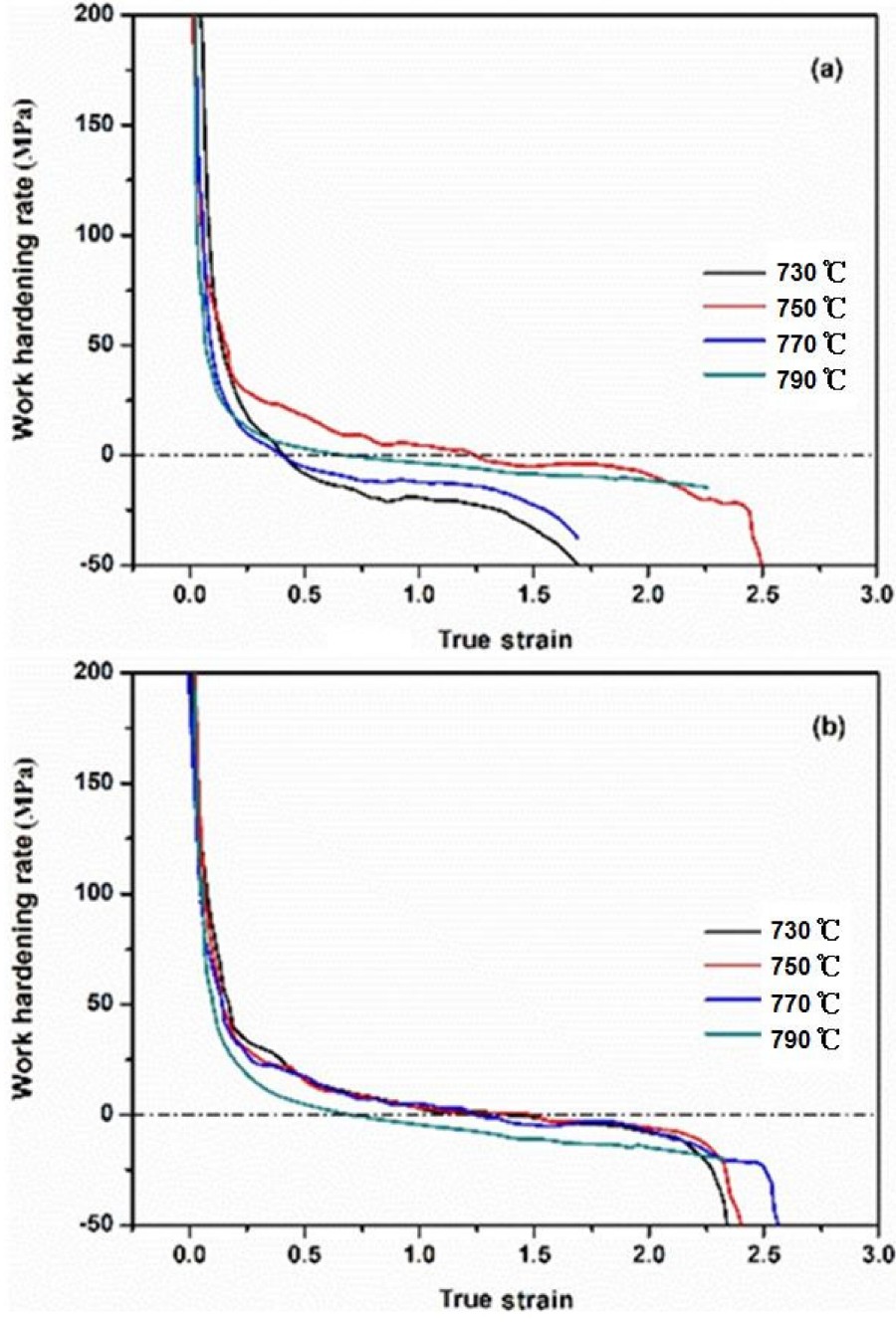

**Figure 6.** The work hardening rate curves of the forged (**a**) SP700 and (**b**) SP700Cr alloys.

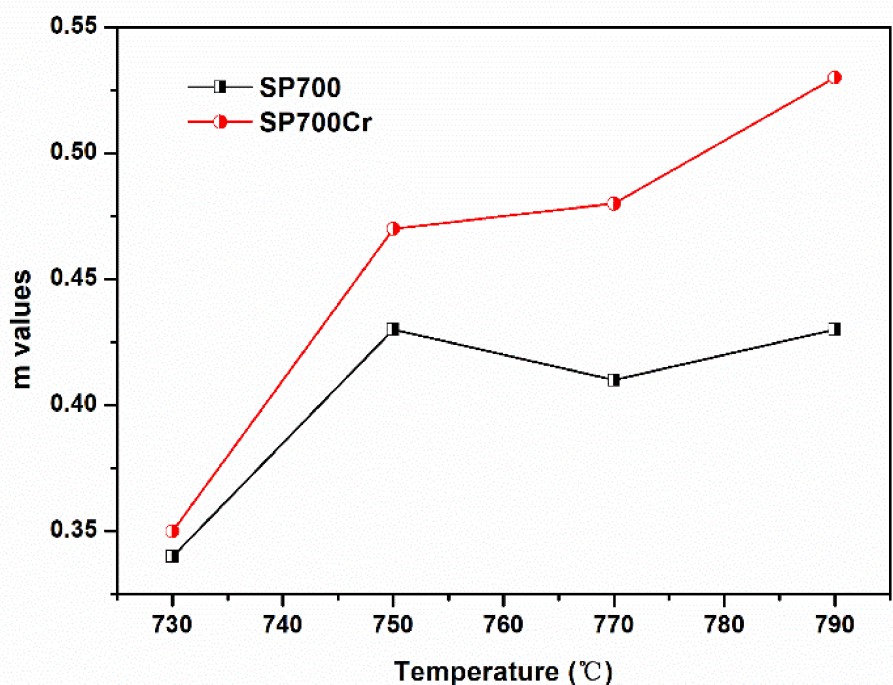

**Figure 7.** Strain rate sensitivity (m) values of the forged SP700 and SP700Cr alloys at different elevated temperatures.

## 4. Discussion

It is well known that the mechanical properties of the alloys are associated with the microstructures, mainly depending on the factors that affect the resistance to dislocation activities. Obviously, the deformation behaviors of both alloys are mainly due to the α and β phases. As compared with that of α phase (hcp crystal structure), β phase (bcc crystal structure) possesses more slip systems and higher diffusivity [5,20,23]. Thus, the deformation occurs more easily as the volume fraction of the β phase increases. As a result, the mechanical properties decrease. After AC710, due to the microstructure of alloy consisting of a large amount of α phase and small amount of β phase, the mechanical properties are a little higher than that of the alloy after AC800 consisting of β phase in the microstructure. When the high density of dispersed β phase precipitates in the alloys after AC820 + AC500, it can effectively hider dislocation motion and result in strain hardening of the alloys, similar to the effect of twins, which leads to the high strength. Meanwhile, the phase/grain boundary not only can extend plastic deformation, but also can act as an effective barrier to dislocation activities. When dislocation activities transfer along phase/grain boundary takes place during deformation, the concentrated stress/strain resulted from dislocation aggregation also easily occur, which would act as the origin of the microcracks. As a result, the plastic compatibility decreases and the early nucleation of crack occurs during the deformation. When the microcracks link with each other or grow into larger crack, plastic instability occurs. The ductility thus dramatically decreases.

To achieve a high plasticity in polycrystalline materials, the basic requirement is a high work hardening ability to resist strain localization and a high deformation accommodation ability to relax stress/strain concentration. For SP700 alloy, the high density of dislocations and twins can be observed in the microstructure of the alloys after superplasticity (Figure 8). These bowed, tangled and homogeneously distributed dislocations indicates that the movement of dislocations is very active in the grain interior regions. These dislocations can interact with other dislocations, interstitial atoms and even the grain boundaries in the slipping process, which can provide tensile strain with increasing dislocation density or an important coordination for grain sliding being good for superplasticity deformation [24]. Meanwhile, twins is also beneficial for the superplasticity due to the fact that it can produce

tensile strain during twining shear, change the orientation of crystals and refine grains in the process of tensile deformation [25,26]. However, due to a weak work hardening ability (Figure 6), the strain localization would occur earlier and the form the origin or nucleation of the microcracks. When these microcracks link with each other, plastic instability occurs thus leading to the fracture. Therefore, the superplasticity of SP700 alloy can be linked to the large number of dislocation activity and twins. For SP700Cr alloy, dislocations with lower density are favored for distribution around the uniform equiaxed grain (mainly β phase) boundaries and the grain/phase boundary trends to be curved and distorted (Figure 8). Due to Cr being a β phase stabilizing element with relatively high diffusivity ($2.04 \times 10^{-9}$ cm$^2$/s at 900 °C [27]) in titanium, the stability of the grain size is good and rapid grain growth or grain elongation would not occur. In order to ensure the continuity of materials during deformation, grains must adjust themselves to adapt to the movement/rotation of neighbor grains [24,28]. The ductile and highly diffusive β phase effectively accommodates grain/phase boundary sliding. Therefore, grain/phase boundary sliding, which is accommodated by the dislocation activities in the regions of grain/phase boundaries, tries to surround grains in order to harmonize the deformation. Meanwhile, the strong work hardening ability (Figure 6) would delay the strain localization and thus the form of microcracks. From this point of view, it is inferred that the grain/phase boundary sliding can improve the adaptability of the grain to deformation, thus improving the superplasticity.

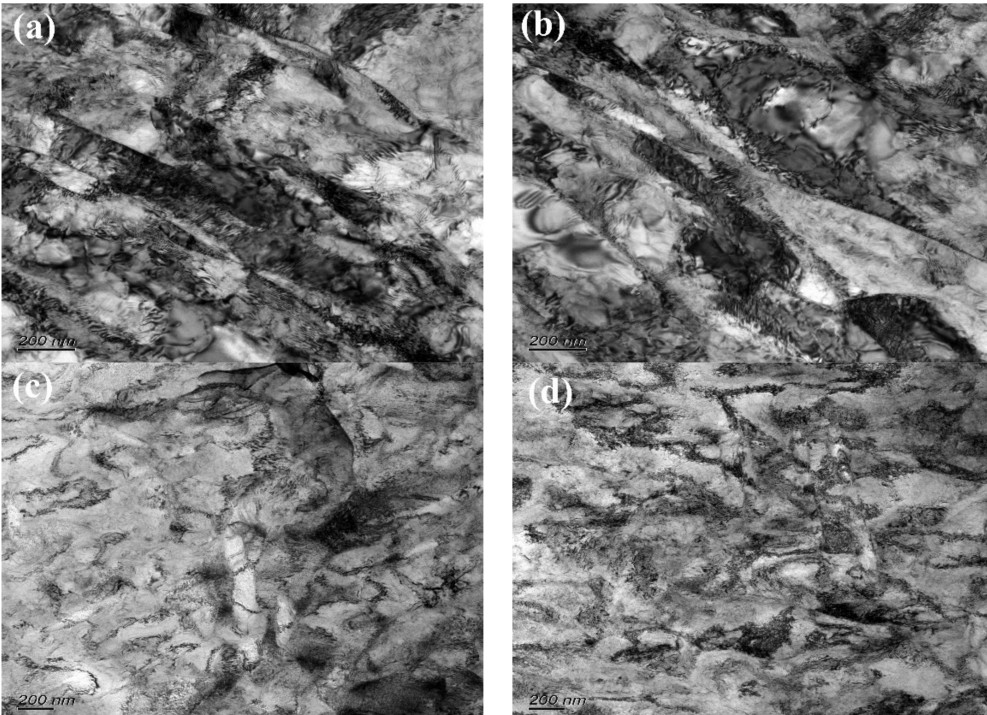

**Figure 8.** Transmission electron microscope (TEM) images of the forged (**a**,**b**) SP700 and (**c**,**d**) SP700Cr alloys after tensile experiment at 770 °C.

In addition, crystal orientation is also an important aspect to understand microstructure characteristics. Figure 9 illustrates the inverse pole figure (IPF) for SP700Cr alloy after AC820 + AC500. The change of crystal orientation of the alpha phase can be observed. In general, the crystal orientation dispersion of α phase is good, and there is no obvious orientation concentration phenomenon, which indicates that the orientation uniformity of α phase is better. However, two typical crystal orientation characteristics can be observed, as shown by the red dashed circle. For grain I, the grain interface is bent and disturbed from the morphological point of view, and there are two low-angle boundaries inside. The grain I show an obvious deformation characteristics. The Euler angles of three subgrains

are calculated as [φ1 = 85.5°, Φ = 77.3°, φ2 = 1.5°], [φ1 = 80.5°, Φ = 78.6°, φ2 = 1.8°], [φ1 = 85.4°, Φ = 71.8°, φ2 = 1.9°]. Their crystal orientations are very close and still belong to one grain. For grain II, crystal orientation inside has shown obvious difference. Their Euler angles are calculated as [φ1 = 113.3°, Φ = 48.2°, φ2 = 24.0°], [φ1 = 117.3°, Φ = 105.8°, φ2 = 45.1°], [φ1 = 113.4°, Φ = 48.2°, φ2 = 25.1°]. From a crystallographic point of view, grain II has actually been separated into three independent grains, although it still belongs to the same grain from a morphological point of view. Such a phenomenon provides a more comprehensive insight for microstructure evolution.

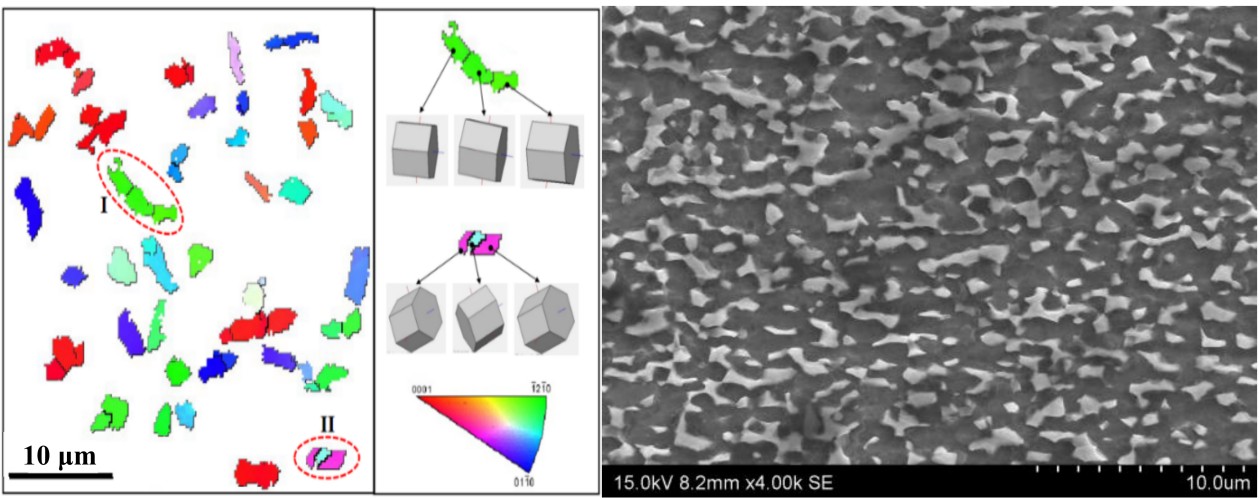

**Figure 9.** The inverse pole figure (IPF) of alpha phase for SP700Cr alloy after AC820 + AC500.

Schmid factor maps for SP700Cr alloy after AC820 + AC500 are shown in Figure 10. From Figure 10, the Schmid factor distribution of basal {0001}, prismatic {1–100} and pyramidal {1–101} slip planes in the closed packed direction <11–20> can be observed. The calculations results indicate that the pyramidal slip is the easiest to activate, prismatic slip comes second, and basal slip is hard to activate. Such behavior can be explained by axial ratio (c/a) of α phase of titanium alloy. For α phase in titanium alloy, axial ratio is about 1.598 which is less than standard value (1.633). In this case, the prismatic or pyramidal slips are easier to activate. The above analysis can help to understand deformation progress of α phase of SP700Cr alloy.

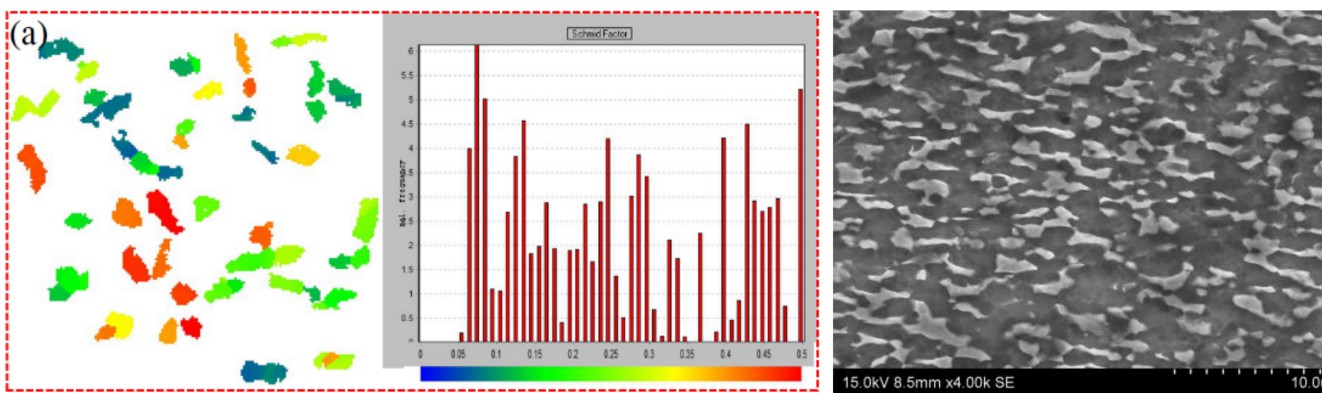

**Figure 10.** *Cont.*

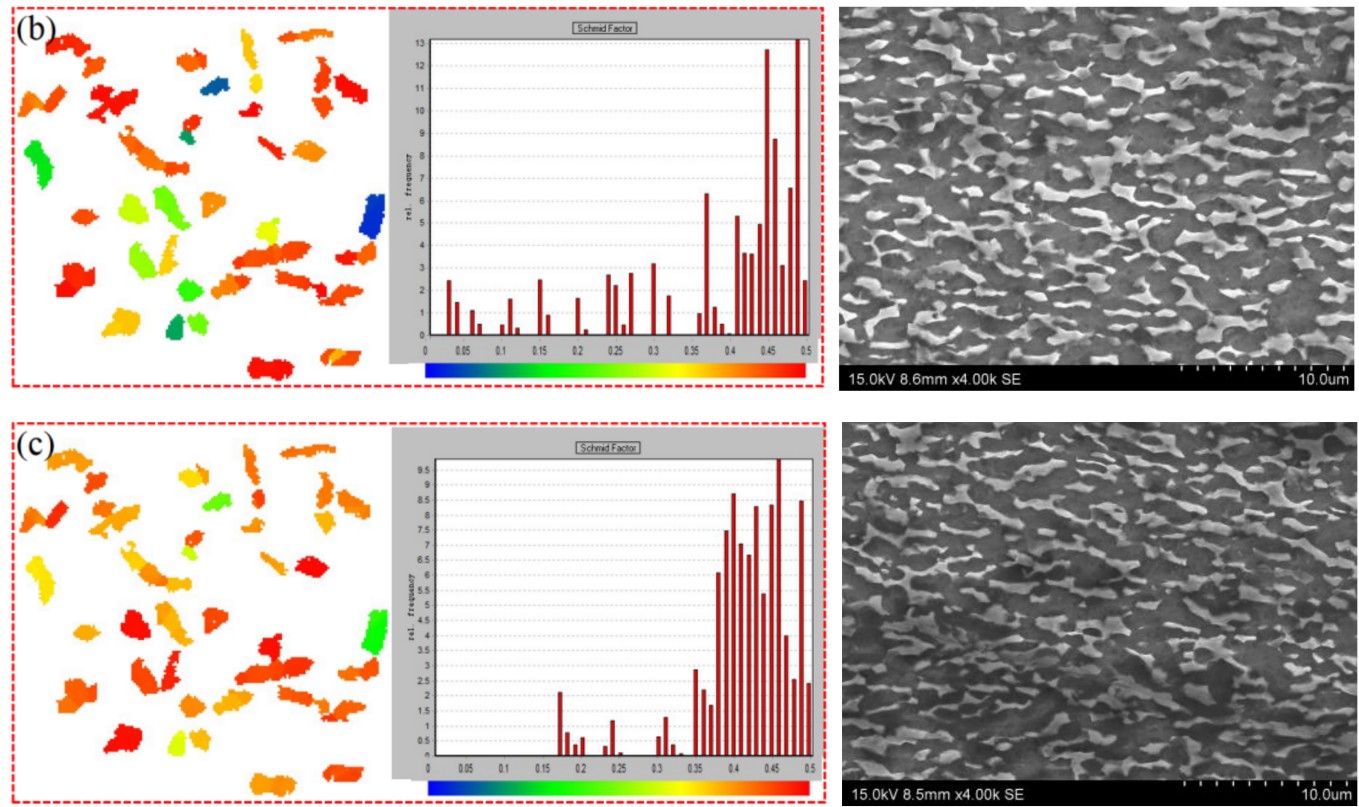

**Figure 10.** Schmid factor maps for SP700Cr alloy after AC820 + AC500: (**a**) {0001}<11–20>, (**b**) {1–100}<11–20> and (**c**) {1–101}<11–20>.

## 5. Conclusions

After annealing at 820 °C for 1 h and aging at 500 °C for 6 h, the strip and interval distributed α phase and β phase were transformed into short or spherical phases. The yield strength, ultimate tensile strength and tensile elongation of SP700 alloy with optimized microstructure was 1312 MPa, 1211 MPa and 10%, and the superplasticity elongation of SP700 alloy with 1.5% Cr addition can reach 1127% at 770 °C. Such properties were attributed to the uniform equiaxed/globular fine microstructure and the superplasticity was attributed to the grain/phase boundary sliding accommodated by the dislocation activities in the vicinity of the grain/phase boundaries. It should be pointed out that the degraded ductility following the AC820 + AC500 treatment may be a limitation for an engineering application, which should be fully considered. The EBSD analysis shows that the orientation uniformity of α phase for SP700Cr alloy after AC820 + AC500 is good. The calculations results of the Schmid factor indicate that the pyramidal slip is the easiest to activate, prismatic slip comes second, and basal slip makes it hard to activate boundaries.

**Author Contributions:** D.H. and J.X. designed the experiments; W.Z. carried out the experiments; D.H. and Y.Z. analyzed the experimental results. D.H. and J.X. prepared the original draft; Y.Z. revised the original manuscript. All authors have read and agreed to the published version of the manuscript.

**Funding:** This research was funded by the National Natural Science Foundation of China (grant No. 52105484), Natural Science Foundation of Shaanxi Province (Grant No. 2021JQ-117), the Fundamental Research Funds for the Central Universities (Grant No. G2020KY0501), Basic Research Programs of Taicang (Grant No. TC2020JC10).

**Institutional Review Board Statement:** Not applicable.

**Informed Consent Statement:** Not applicable.

**Data Availability Statement:** The data presented in this study are available on request from the corresponding author.

**Conflicts of Interest:** The authors declare no conflict of interest.

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
