# Peer review of "Investigation of Slow Eutectoid Element on Tensile Properties and Superplasticity of a Forged SP700 Titanium Alloy"

_metals, doi:10.3390/met11111852_

Round 1

Reviewer 1 Report

In the manuscript «Investigation of slow eutectoid element on tensile properties and super-plasticity of a forged SP700 titanium alloy », the authors investigated the mechanical properties and superplasticity of the forged SP700 alloy without and with the addition of a slow eutectoid element (1.5%Cr). The authors carried out detailed studies of the effect of chromium on and thermomechanical treatment on the evolution of the microstructure, superplasticity, and mechanical properties of the SP alloy.

I have the following comments on the manuscript:

  1. Title. “Investigation of Slow Eutectoid Element on Tensile Properties and Super-Plasticity of a Forged SP700 Titanium Alloy”. Must be “Superplasticity”.
  2. Materials and experimental procedures. First paragraph: «Materials and experimental procedures The Materials and Methods should be described with sufficient details…». Please remove it.
  3. Figure 3. Please add an error bar for strength characteristics. This is important for evaluating differences in properties.
  4. 3.2 Tensile and super-plastic properties. Must be “Superplasticity”.
  5. Figure 6. “The work hardening rate curves…”. Specify the units of measure for the strain hardening rate.
  6. 4. Discussion. “Due to Cr is a β phase stabilizing element with high diffusivity, the stability of the grain size…”. Why do the authors talk about the high diffusivity of chromium in titanium and at the same time do not identify the values of the diffusion coefficient? There are many publications with diffusivities for many elements in titanium.
  7. For superplastic behavior, grain size is of main importance. Why didn’t the authors provide a grain size comparison? This is important in terms of the effect of chromium and heat treatment.

Author Response

Please see the point-by-point response to the reviewer’s comments in the attachment.

Reviewer 2 Report

The reference alloy SP700 is not properly defined. Is this alloy a standard product (in this case the reference to the applicable standard should be mentioned), or it is the proprietary alloy “JFE SP-700” (not “SP700”) from JFE Holdings, Inc., Japan?

The authors studied a forged products. No details are provided about the starting product (in particular, form and size, temper state of the forging stock etc.). No details are provided about the applied forging process (multidirectional redundant forging?) and schedule, in particular the total reduction ratio. The shape and size of the final forging is not reported.

Table 1 reports a chemical composition (product analysis or heat analysis?) which has certainly not been measured. Indeed, all reported elements for SP700 are exactly in the middle of the allowed respective ranges for the JFE SP-700 proprietary product. For the SP700 Cr product, same applies except for Fe and Cr which have been voluntarily modified. Moreover, key elements to be guaranteed by specification are not measured and reported, namely C, H, N, O, Y. Their role on the achieved properties is nowhere discussed in the paper.

In the heat treatment schedule, a “No” heat treatment temper appears (as forged?). The results of mechanical properties for this temper are not reported in Fig. 3a or 3b.

It is not mentioned according to which standard the tensile properties have been carried out.

It is mentioned that “The morphologies of the fracture surfaces after super-plastic deformation were ob-served by the transmission electron microscope (TEM)”. TEM is not usable to study the morphology of a broken surface, it is SEM which is the adapted tool. Consistently with this remark, later on, no images of the fracture surfaces are provided. Only lamellae have been TEM observed, close to the fracture surface?

The results provided if Fig. 3 are very poor in terms of ductility for all tempers and for both alloys. For reference, the ductility of annealed JFE SP-700 under the form of bars, 25 mm diam., is 21 % and no temper reached this value. The AC820+AC500 temper in particular, features low ductility in the order of 10 %.

Moreover, it is not reported in which direction the properties have been tested. In particular, Ti-alloy forgings if not properly processed, can feature very poor ductility in the short transverse direction. Have the properties been conservatively assessed for this direction?

Finally, the writing of this paper is very neglected and deeply under the level of minimum requirements for a scientific publication. The y-axis of Fig. 6a and 6b have no units. The x axis of Fig. 5a, 5b, 6a, 6b mention a “ture” stress. English is hardly readable, with extensive grammar and syntax errors.

Author Response

(The authors gave the same response as above.)

Reviewer 3 Report

Dear authors,

I have some suggestions in order to improve your manuscript:

  1. Please delete the first paragraph in chapter 2, I think it is not a part of your paper "The Materials and Methods should be described with sufficient details to allow others to replicate and build on the published results....."
  2. Please provide a clearly presentation in Table 2 of the last heat treatment schedule. It is not clear 820C for 1h, then AC, then ?... 
  3. For Fig.1 and Fig.2 please provide a more visible (bigger) scale bar on optical images. 
  4. In Fig. 3 I suggest you to convert the data presented as a graph into a table since only 3 heat treatments were performed 
  5. For Fig.9 and Fig 10 I suggest you to present also the correspondent morphological area 

Thank You

Author Response

(The authors gave the same response as above.)

Round 2

Reviewer 1 Report

The authors revised the manuscript in accordance with the comments and answered questions regarding the results. I think that now the manuscript can be accepted for publication.  

Author Response

Thank you very much.

Reviewer 2 Report

The responses of the authors are not fully satisfactory.

  • Concerning the reduction ratio, it is answered that “The selected sample is a cylindrical sample, with a diameter of 80mm and a height of 1200mm. It is forged to 96mm with compression reduction of 20%. The final sample is a cylinder with a diameter of 88.9 ~ 95.2mm with middle bulge. “

This reduction of 20 % is a negligible one, moreover in a single direction, for a product to be considered as forged. A minimum reduction ratio in the order of at least 3 or 6 should have been considered, and a redundant forging should have been applied. By the way, it is unclear how a product of a height of 1200 mm (?) and a diameter of 80 mm can have been upset in compression down to a diameter of 96 mm, the aspect ratio of the product does not allow that. Finally, still no details are provided about the forging temperature etc. Moreover, this information has not been added to the text.

  • More in general, the text has not been completed (e.g. the reference standard for tensile tests) in function of the responses provided.
  • Table 1 reporting the chemical composition is still named “Actual composition”, but as confirmed by the response of the authors this is neither a heat nor a product analysis issued from certificates or measurements. The Table just reports the mid-range of the nominal composition, and even in this respect it is incomplete. The composition of the products studied is not sufficiently characterized.
  • The degraded ductility following the AC820+AC500 treatment is an important limitation for an engineering application of the alloys in this temper, moreover it has been measured in a favourable (longitudinal) direction. The present version of the paper still does not address/comment this limitation, on the contrary it concludes on “excellent tensile properties” without any caveat or limitation.
  • The English is still not acceptable and several expressions are incomprehensible (“the lower density of dislocations are favor to distribute in SP700Cr alloy” etc.). Spelling (“super-plascity”; “phographic”) and grammar should have been thoroughly checked as suggested in the first revision, which has not been the case.

Round 3

Reviewer 2 Report

The authors should at least provide in Table 1 the composition as reported in the mill certificates from the suppliers of the two products of the respective alloys which have been used as pre-material for the forgings. These certificates containing the information on composition (at least the heat analysis) should in any case be available with the authors to confirm the proper identification and traceability of the products used for the present study.    

Author Response

Thank you very much for the valuable comments. According to your suggestion, we give the detailed information on composition in Table 1.